# Comparative Transcriptome Analysis Reveals Potential Genes Conferring Resistance or Susceptibility to Bacterial Canker in Tomato

**Shuozhen Deng, Ziyan Li, Xinyu Liu, Wencai Yang and Yuqing Wang \***

Department of Vegetable Science, College of Horticulture, China Agricultural University, Beijing 100193, China
* Correspondence: wyq@cau.edu.cn

**Abstract:** Bacterial canker of tomato is a systemic disease caused by *Clavibacter michiganensis* (*Cm*), which poses a grave threat to tomato production worldwide. Towards the identification of genes underlying resistance to *Cm* infection, the transcriptome of the resistant inbred backcross line IBL2353 carrying the *Rcm2.0* locus derived from *Solanum habrochaites* LA407 and the susceptible *Solanum lycopersicum* line Ohio88119 was comparatively analyzed after *Cm* inoculation, and the analysis focused on the genes with different expression patterns between resistant and susceptible lines. Gene ontology (GO) analysis revealed that top terms of differentially expressed genes comprised ubiquitin protein ligases, transcription factors, and receptor kinases. Then we screened out some genes which are potentially associated with the defense response against *Cm* infection in IBL2353 including the wall-associated receptor kinase-like 20 (*WAKL20*), and virus-induced gene silencing showed it contributes resistance to *Cm* infection. In addition to *Cm*-induced genes related to resistance, the expression of eight homologs from six susceptibility (*S*) gene families was analyzed. These putative resistance and susceptibility genes are valuable resources for molecular resistance breeding and contribute to the development of new control methods in tomato.

**Keywords:** bacterial canker; RNA-seq; comparative transcriptome; inbred backcross line

## 1. Introduction

Bacterial canker in tomato is one of the most damaging diseases which is caused by the Gram-positive bacterium *Clavibacter michiganensis* (*Cm*) [1–3]. Since its first occurrence in Michigan, USA in 1909 [4], this disease has been found in almost all tomato production areas in more than 80 countries in Asia, Europe, Africa, America, and Oceania [2,5]. It causes severe economic losses varying from 10% to 100%. However, there is an absence of powerful methods for controlling bacterial canker in tomato [3,6].

Bacterial canker of tomato is a systemic vascular disease which can emerge at all growth stages of tomatoes, and the pathogen can invade tomato plants through natural entries and wounds such as hydathodes, stomata, and trichomes [7]. The typical symptoms are cankers on stems, the unilateral wilting of leaves, and the appearance of bird-eye spots in infected fruits [8–11]. Because the pathogen of this disease mainly spreads and propagates in the host's interior vascular bundles, it is difficult to control through chemical or integrated management. Although, people have identified some wild resistant accessions to bacterial canker, and have still not cloned the resistant genes and transferred the resistance into cultivars [3,8,12]. So far, *S. habrochaites* LA407 and *S. arcanum* LA2157 are two of the most studied *Cm*-resistant accessions, and several QTLs have been identified in them, respectively [3].

Genome-wide transcriptome analysis is a powerful way to find out the host molecular responses to pathogen infection. Herein, we have performed a comparative transcriptome analysis using RNA-seq to disclose the important *Cm* resistance molecular players in the tomato. Revealing molecular basis of the *Cm* infection response in tomato largely counts

on the transcriptomic discrepancies between the susceptible and resistant genotypes, preliminary to and after bacterial pathogen inoculation. Formerly, thousands of differentially expressed genes (DEGs) were identified in transcriptome analysis between susceptible cultivars and resistant wild *S. arcanum* LA2157 [13–15]. Of interest, 122 receptor-like kinases participated in pattern-triggered immunity (PTI) and 46 transcription factors in susceptible tomato cultivars [13]. The proteome-level analysis of *Cm*-inoculated tomatoes disclosed a series of differentially expressed PR proteins [16]. In spite of such understanding on host–pathogen interactions in tomato, the understanding of the defense mechanism was still incomprehensible, since there were too many DEGs to analyze related to the complicated pathophysiology interaction. Therefore, we envisaged that a comparative transcriptome with less background difference between two lines may bridge the prevalent weakness and broaden our current knowledge of the complicated interaction and impact on the outcome of infection.

In the past several decades, researchers have focused on developing the dominant resistance (R) genes from resistant accessions, whose products mediate the specific pathogen strains' recognition and protection [17]. Nevertheless, resistance mediated by the dominant *R* gene is readily broken by the escape or mutations of effectors in pathogens for survival evolution. Susceptibility (S) genes are, in contrast, highly resistant to evolutionary change as these are typically recessive, hence the gradual change in focus on *S* gene research in recent studies. The inactivation of *S* genes is more likely to create durable and broad-spectrum resistance in crops [18], and *S* genes are usually conserved among plant species [19]. In the present study, we analyzed 28 *S* gene orthologs [18,19] potentially controlling the susceptibility of bacterial canker and provided insights that may contribute to the strategies controlling bacterial canker of tomato.

Based on the above background, we utilized one inbred backcross line IBL2353 which stemmed from *S. habrochaites* LA407 as the resistant object in comparative analysis. IBL2353 was identified as that maintaining resistance sources in a genetic background with lower than 4.2% of the LA407 genome, which overcame the low mapping percentages and large genetic variation between cultivars and wild materials [20]. Quantitative reverse transcription PCR (RT-qPCR) and transient gene-silencing experiments were designed to identify and correlate some specifically induced genes to the defense response after *Cm* infection in two tomato lines. Our results present a series of potential defense-related candidate genes in tomato–*Cm* interaction, which will contribute to better understanding the molecular basis of resistance against bacterial canker and the next resistant gene utilization in tomato breeding.

## 2. Results

### 2.1. Phenotype Response of IBL2353 and Ohio88119 to Cm Infection

Healthy plants with 5–6 true leaves were inoculated with *Cm* bacterial suspension as treated samples and with MgSO$_4$ solution as mock samples. Ohio88119 mock-inoculated plants were free of symptoms at 30 days post-inoculation (dpi) (Figure 1A). Ohio88119 infected plants showed symptoms as early as 15 dpi including necrotic lesions at the leaves' edges and the wilting of mature leaves. Upon continued incubation at 25–30 dpi, the typical canker symptoms appeared. The symptoms included the unilateral wilting of compound leaves and unilateral plants wilting finally; long and cracking injection sites of the stems; and the dying of whole plants at 30 dpi (Figure 1B–D). In contrast, IBL2353 infected plants remained symptom-free at 30 dpi and showed no apparent differences to the mock-inoculated plants (Figure 1E,F). Some of the IBL2353 infected plants just displayed small and mild canker wounds located in the inoculation site and occasionally wilting leaves at 30 dpi (Figure 1F,G). Therefore, the phenotype of inbreeding line IBL 2353 was very similar to the control plants after *Cm* infection and confirmed the resistance to the *Cm* pathogen. In conclusion, both tomato lines appeared the expected phenotype difference upon *Cm* infection.

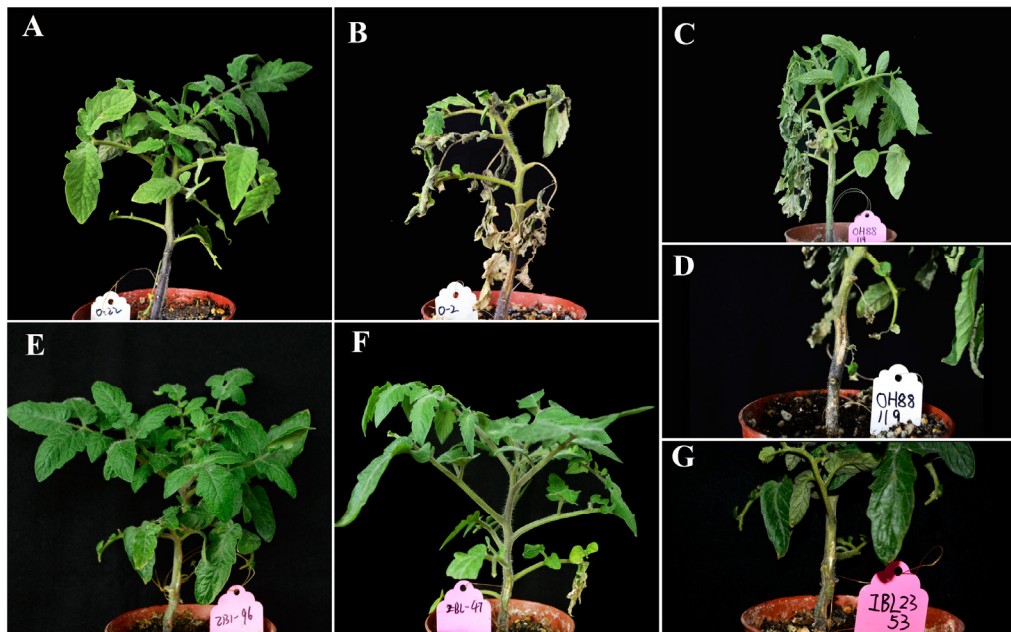

**Figure 1.** Symptoms of bacterial canker in resistant inbreeding line IBL2353 and susceptible *S. lycopersicum* Ohio88119 infected with the *Cm* GS12102 strain. (**A**) Ohio88119 plant at 30 days after mock inoculation. (**B**) Ohio88119 plant showed the wilting of whole plant at 30 days after *Cm* inoculation. (**C**) Unilateral wilting of the whole plant in Ohio88119 at 25 days after *Cm* inoculation. (**D**) Long and cracking canker on the injection sites of susceptible line at 30 days after *Cm* inoculation. (**E**) IBL2353 plant at 30 days after mock inoculation. (**F**) IBL2353 plant with mild leaflet wilting at 30 days after *Cm* inoculation. (**G**) Small and mild canker wound on the injection sites of resistant line at 30 days after *Cm* inoculation.

### 2.2. Cm-Induced Differentially Expressed Genes

Transcriptome profiling was conducted with both resistant (IBL2353) and susceptible (Ohio88119) genotypes at 0, 12, and 24 h after *Cm* infection. All RNA samples showed high Q30 quality scores (Table S1). The average of mapping percentages in IBL2353 at three time-points was 95.0%, while the average in Ohio88119 was 96.0% (Table S2). DESeq statistical analyses identified differentially expressed transcripts with a fold-change >2.0 and false discovery rate (FDR) ≤ 0.05. The overlap in DEGs at two time-points is shown in Supplementary Figure S1. Of the up-regulated DEGs identified, about 50% of the 12 h samples and about 32% of the 24 h samples were in common in the IBL2353 and Ohio88119 (Figure 2A). The down-regulated DEGs were 54% in common from the 12 h samples (Figure 2B).

Excluding the overlapped DEGs in the resistant line and the susceptible line, we concentrated on those DEGs with different expression patterns between two lines after *Cm* infection. Then, 1130 up-regulated DEGs were screened out in the resistant IBL2353 (the sum of black rectangles in Figure 2C), and another 118 genes appeared delayed up-regulation in Ohio88119 at 24 hpi (the sum of the black circles in Figure 2C). In IBL2353, 907 genes, marked with three blue rectangles, were specifically down-regulated (Figure 2D) and 814 genes, marked with three red rectangles in Figure 2C, were up-regulated uniquely in the susceptible line without a change in the resistant line.

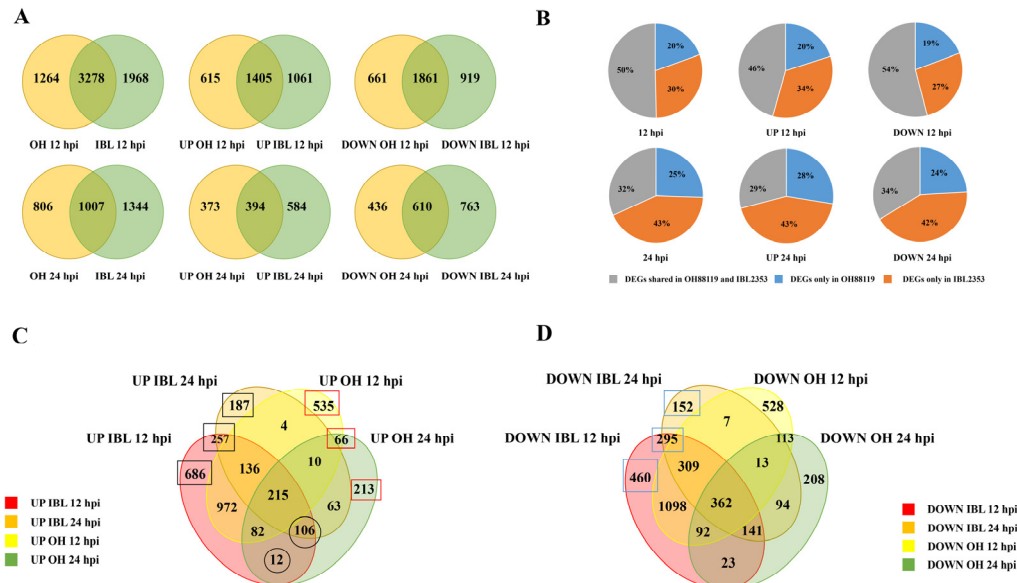

**Figure 2.** Venn diagrams showing DEG contrasts between resistant IBL2353 and susceptible Ohio88119. (**A**) Venn diagram representation of overlapping DEGs quantity between two tomato lines. (**B**) The pie chart representation of the proportion of DEGs shared in Ohio88119 and IBL2353. (**C**) Comparison of up-regulated DEGs between IBL2353 and Ohio88119 at 12 and 24 hpi. (**D**) Comparison of down-regulated DEGs between IBL2353 and Ohio88119 at 12 and 24 hpi. UP and DOWN herein are the abbreviation of up-regulated and down-regulated, and IBL and OH herein are the abbreviation of IBL2353 and Ohio88119.

### 2.3. GO Term Enrichment and KEGG Pathway Analysis

The DEGs with different expression patterns between two lines (Figure 2C,D) were used for gene ontology (GO) terms and KEGG analysis. Tables S3–S5 shows the enriched GO categories: biological process (BP), molecular function (MF) and cellular component (CC). The top 20 enriched GO terms are listed as shown in Figure 3A–C. The BP category contains genes related to the positive regulation of ubiquitin protein ligase activity, DNA replication initiation, the carbohydrate metabolic process, and the alcohol metabolic process, which have higher enrichment than other processes. Ubiquitin-protein transferase activator activity, transcription factor activity (sequence-specific DNA binding), anaphase-promoting complex binding, and protein disulfide oxidoreductase activity were the top four MFs in the degree of enrichment. Moreover, the transcription factor activity (sequence-specific DNA binding) had the highest number of genes in IBL2353 during the 24 h after *Cm* infection. In CC terms, MCM complex, nucleosome, host cell nucleus, and integral component of membrane were top four terms.

The identified DEGs were mapped to the KEGG database to obtain an insight into the major metabolic pathways operating in response to *Cm* infection. The pathway enrichment analysis assigned a KEGG number to 2155 DEGs and mapped them into 123 pathways in the resistant IBL2353 (Table S6). The top 20 pathways in connection with these DEGs are shown in Figure 3D. Among them, the biosynthesis of globo and isoglobo series glycosphingolipid was the most enriched pathway. Overall, the GO and KEGG analyses were in line with the putative role of the identified DEGs in the immune responses of tomato against *Cm* pathogen infection.

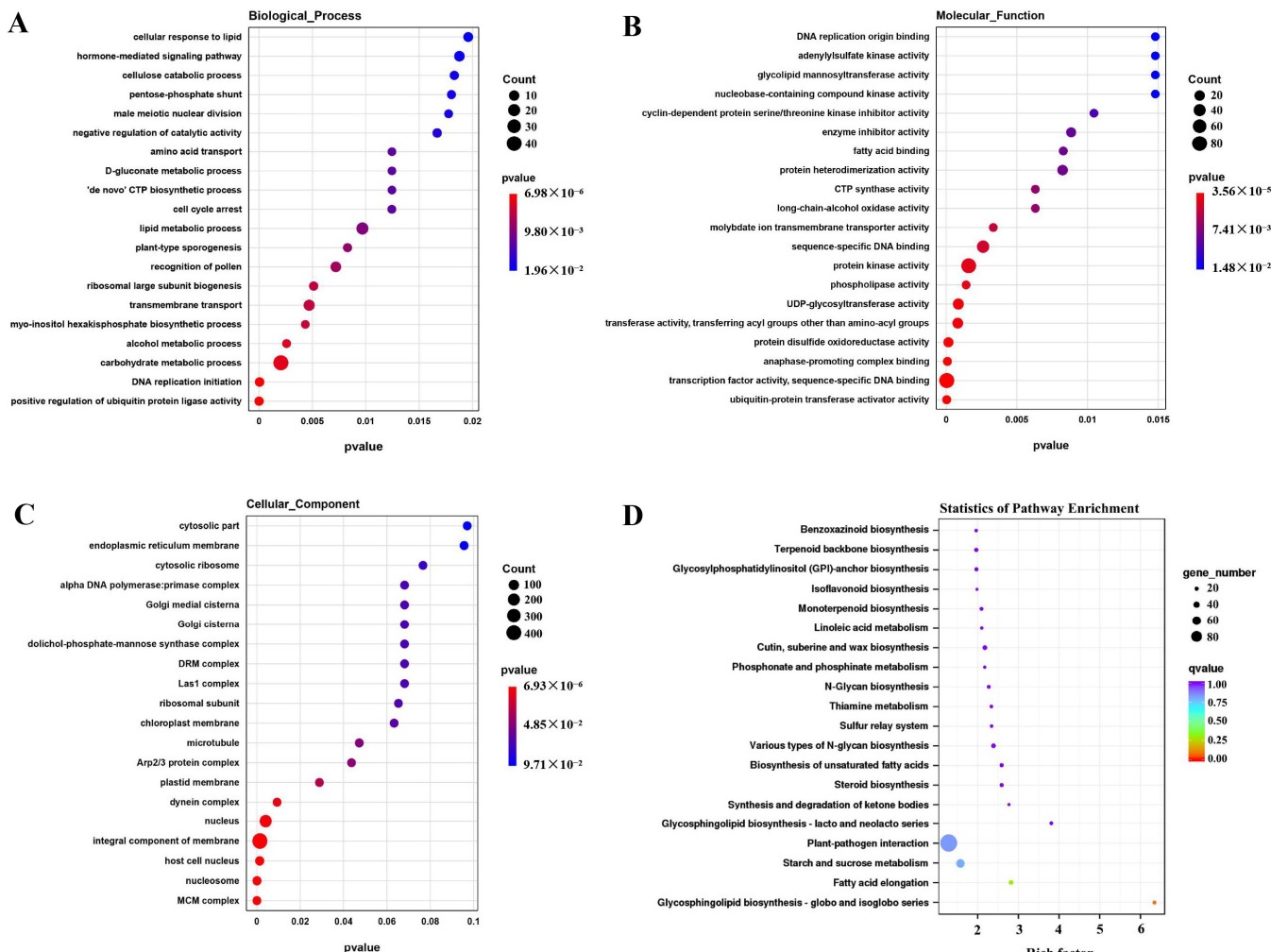

**Figure 3.** GO enrichment and KEGG pathway analysis of genes with different change patterns in two lines. (**A**) shows the top 20 enriched GO terms on Biological Process. (**B**) shows the top 20 enriched GO terms on Molecular Function. (**C**) shows the top 20 enriched GO terms on Cellular Component. The size of the circles represents the number of DEGs annotated in this pathway, and the color of the circles represents the *p* value. A lower *p* value indicates greater enrichment intensity. (**D**) Scatterplot of KEGG pathways shows the top 20 enriched pathway terms in genes with different change patterns in two lines. The sizes of the circles indicate the number of genes. The Rich factor is the ratio of the number of DEGs annotated in a given pathway to the number of all genes annotated in this pathway. The q value is the corrected *p* value and ranges from 0 to 1, with a lower q value indicating greater intensity.

### 2.4. Genes Associated with the Defense Response against Cm Infection

Combining the DEGs values (log fold change (FC) ≥ 1) with the GO term and KEGG pathway enrichment analyses and the gene functional annotation associated with resistance, 25 genes were selected for further analysis. To compare the differences in these 25 genes between the resistant and susceptible genotypes, a heat map exhibiting the expression profiles was generated (Figure 4). It shows that 11 genes were highly induced at 12 dpi and then dropped back to the 0 h expression level at 24 hpi. This cluster includes the genes coding for an ERAD-associated E3 ubiquitin-protein ligase HRD1B (Solyc03g096930), a probable CCR4-associated factor 1 (CAF1) homolog 11 (Solyc01g007840), several receptors, and receptor-like protein kinases (RLKs). Another 12 genes were up-regulated continuously during the 12 and 24 h after infection in IBL2353. This cluster includes the genes coding for two E3 ubiquitin-protein ligase (MPSR1 Solyc05g007895 and UPL5-like

Solyc10g083470), two LRR receptor-like protein kinases, a wall-associated receptor kinase-like 20 (Solyc09g008640), a disease resistance protein RPP13-like, and six RLKs. There were also two genes expressed most highly at 24 hpi in IBL2353, which were a NDR1/HIN1-like (NHL) protein 6 (Solyc12g095980) and a putative disease resistance RPP13-like protein 3 (Solyc04g009130), respectively. In sum, 14 RLKs were induced to be up-regulated in resistant IBL2353 and probably triggered a PTI response against the *Cm* pathogen.

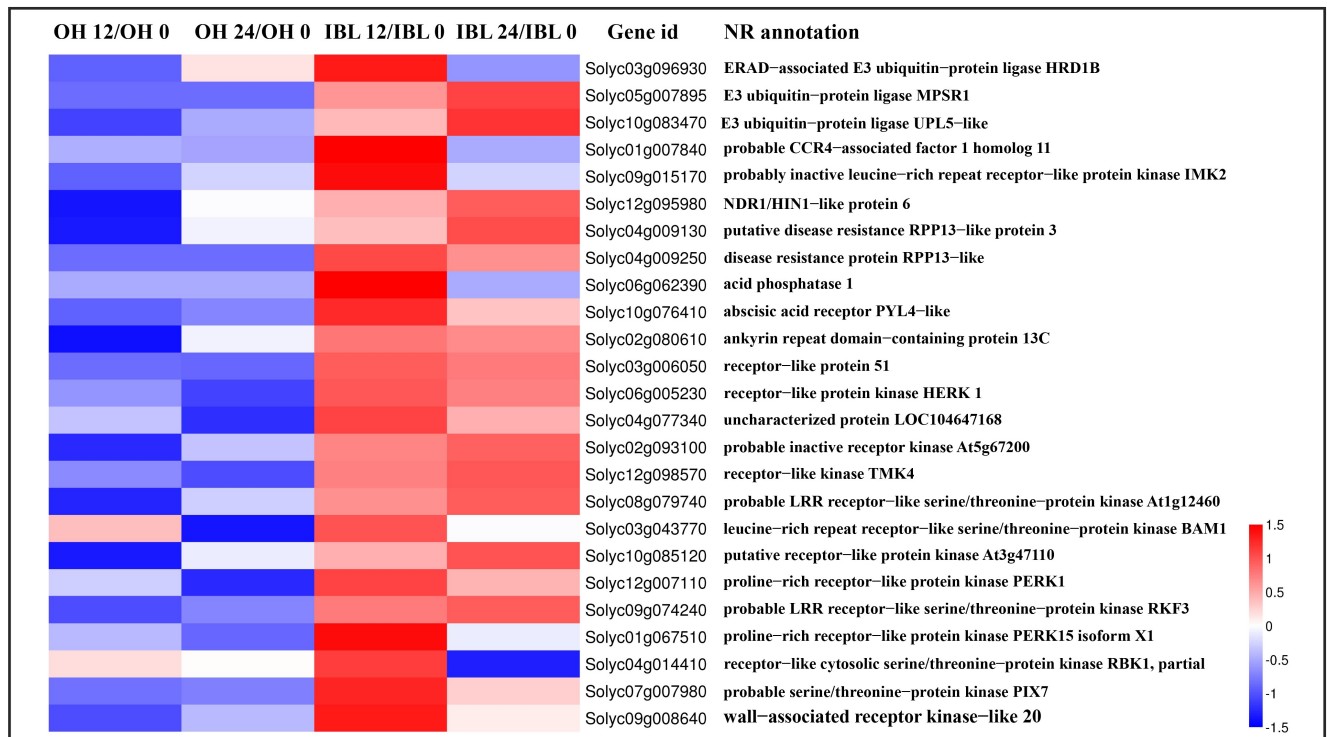

**Figure 4.** Heat maps exhibiting the expression profiles of potentially resistance-related genes. Data were the Log2 value of gene expression fold changes and were normalized based on Z-score formula. The blue color means the value was lower than the mean of 4 Log2 values from two lines at two time-points, and not the down-regulation. The red color means the value was higher than the mean of 4 Log2 values from two lines at two time-points, but not up-regulation. The darker color represents the higher deviation from the average. OH 12/OH 0 and OH 24/OH 0 represent the ratio of gene expression value at 12 hpi and 24 hpi compared to value at 0 hpi in Ohio88119, respectively; IBL 12/IBL 0 and IBL 24/IBL 0 represent the ratio of gene expression value at 12 hpi and 24 hpi compared to value at 0 hpi in IBL2353, respectively.

Besides the above genes, another DEGs category associated with the defense response to the *Cm* pathogen were transcription factors (TFs). Candidate TFs were found to belong to 17 families including 57 TFs (Figure 5). There were 22 TFs up-regulated specifically in IBL2353 at both time-points, belong to 9 different TF classes (Figure 5A), while 35 genes were significantly down-regulated in IBL2353 at both 12 and 24 hpi (Figure 5B), mostly belonging to the two TF families WRKY and AP2/ERF-ERF. Since these TFs specially changed in the resistant line, we speculated that they were involved in the response to *Cm* infection.

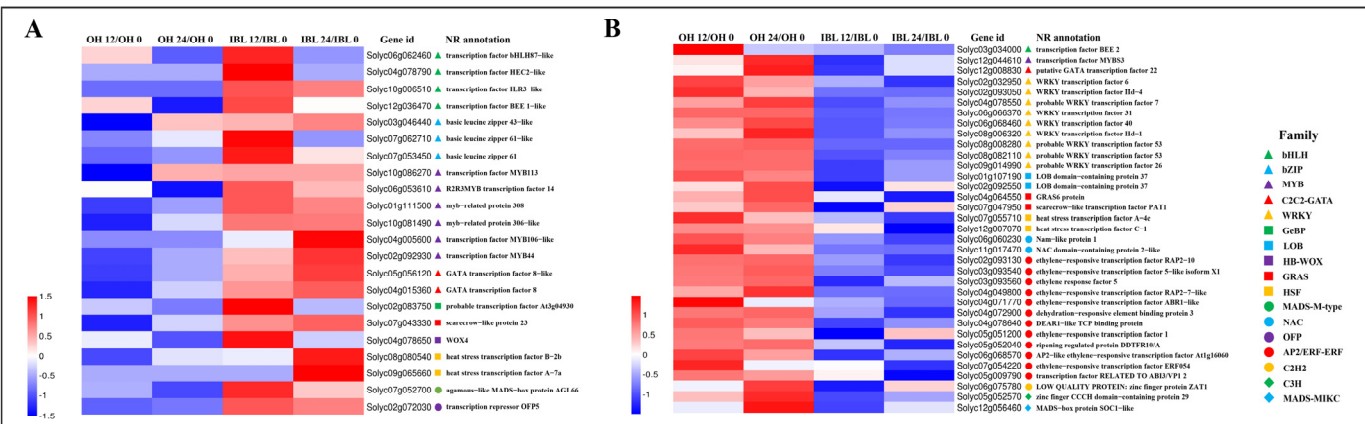

**Figure 5.** Heat maps exhibiting the expression profiles of resistance-related differentially expressed-transcription factor genes. (**A**) TFs up-regulated in IBL2353. (**B**) TFs down-regulated in IBL2353. Data were the Log2 value of gene expression fold changes and were normalized based on Z-score formula. The blue color means the value was lower than the mean of 4 Log2 values from two lines at two time-points, and not the down-regulation. The red color means the value was higher than the mean of 4 Log2 values from two lines at two time-points, but not up-regulation. The darker color represents the higher deviation from the average. OH 12/OH 0 and OH 24/OH 0 represent the ratio of gene expression value at 12 hpi and 24 hpi compared to value at 0 hpi in Ohio88119, respectively; IBL 12/IBL 0 and IBL 24/IBL 0 represent the ratio of gene expression value at 12 hpi and 24 hpi compared to value at 0 hpi in IBL2353, respectively.

## 2.5. Candidate Susceptibility Genes

To screen for putative *S* genes in response to *Cm* infection, genes were compared to previously reported 28 *S* genes (Table S7). Based on the Log2 FC value, eight *S* gene homologs were retained for analysis (Figure 6A). These homologs belonged to six different *S* gene families, and were all induced significant up-regulation in the susceptible line (Figure 6B), including two serine/threonine-protein kinase PBL9 and PBL3 isoform X1 of BIK1 homologs, two LOB domain-containing protein 37 of LOB homologs, a protein-tyrosine-phosphatase MKP1-like of MKP1 homolog, a cyclic nucleotide-gated ion channel 4-like of DND1 homolog, and a glutamate decarboxylase of the GAD homolog. In addition, one putative lipid-transfer protein DIR1 of LTP3 homolog was significantly up-regulated at 24 hpi in the Ohio88119. The GAD homolog (Solyc03g098240) was also up-regulated in the IBL2353 but not to a significant level (Log2 FC $\geq$ 1).

## 2.6. Validation of RNA-Seq by RT-qPCR

Quantitative RT-PCR was carried out to verify the expression patterns of some candidate genes as identified through the RNA-Seq transcriptome analysis. Ten DEGs of potential interest, showing different expression patterns between the resistant and susceptible lines, were selected for validation. Eight of them were potentially related to resistance against *Cm* in the resistant line, which included four transcript factors bHLHs (Solyc12g036470 and Solyc06g062460), bZIP (Solyc07g062710), MYB (Solyc01g111500) and two RLKs (Solyc09g008640 and Solyc08g079740), a putative disease resistance RPP13-like protein 3 (Solyc04g009130), and an abscisic acid receptor PYL4-like (Solyc10g076410). Another two DEGs were *S* genes LOB domain-containing 37 (Solyc01g107190 and Solyc02g092250) induced in the susceptible line. Although there was a minor difference in the fold change of expression compared to RNA-seq, the RT-qPCR results exhibited high consistency with the RNA-Seq data and set up the reliability of transcriptome sequencing conducted in this study (Figure 7).

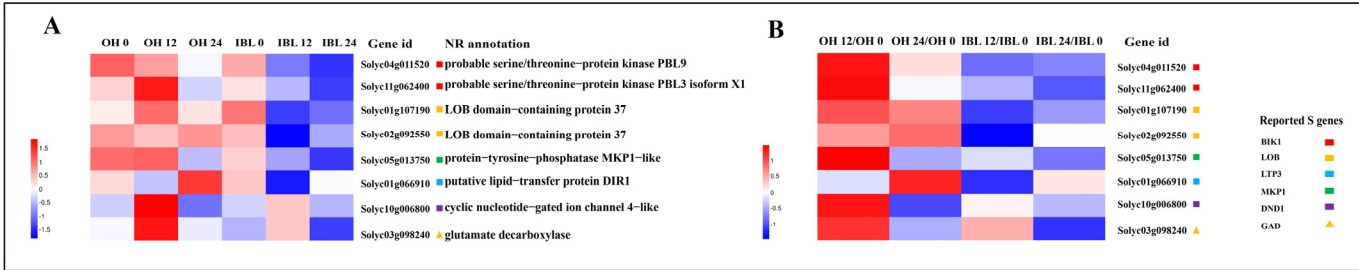

**Figure 6.** Heat map profiles of S gene homologs. (**A**) The quantitative analysis of gene expression at three time-points in two lines. OH 0, OH 12, and OH 24 represent the normalized expression values at 0 hpi, 12 hpi, and 24 hpi in Ohio88119; IBL 0, IBL 12, and IBL 24 represent the normalized expression values at 0 hpi, 12 hpi, and 24 hpi in IBL2353. (**B**) The contrasts of gene expression pattern in two lines. Data were the Log2 value of gene expression fold changes and were normalized based on Z-score formula. The blue color means the value was lower than the mean of the same row of four values from two lines at different time-points, and not the down-regulation. The red color represented the value was higher than the mean of the same row of four values from two lines at different time-points, but not up-regulation. The darker color represents the higher deviation from the average. OH 12/OH 0 and OH 24/OH 0 represent the ratio of gene expression value at 12 hpi and 24 hpi compared to at 0 hpi in Ohio88119, respectively; IBL 12/IBL 0 and IBL 24/IBL 0 represent the ratio of gene expression value at 12 hpi and 24 hpi compared to value at 0 hpi in IBL2353, respectively.

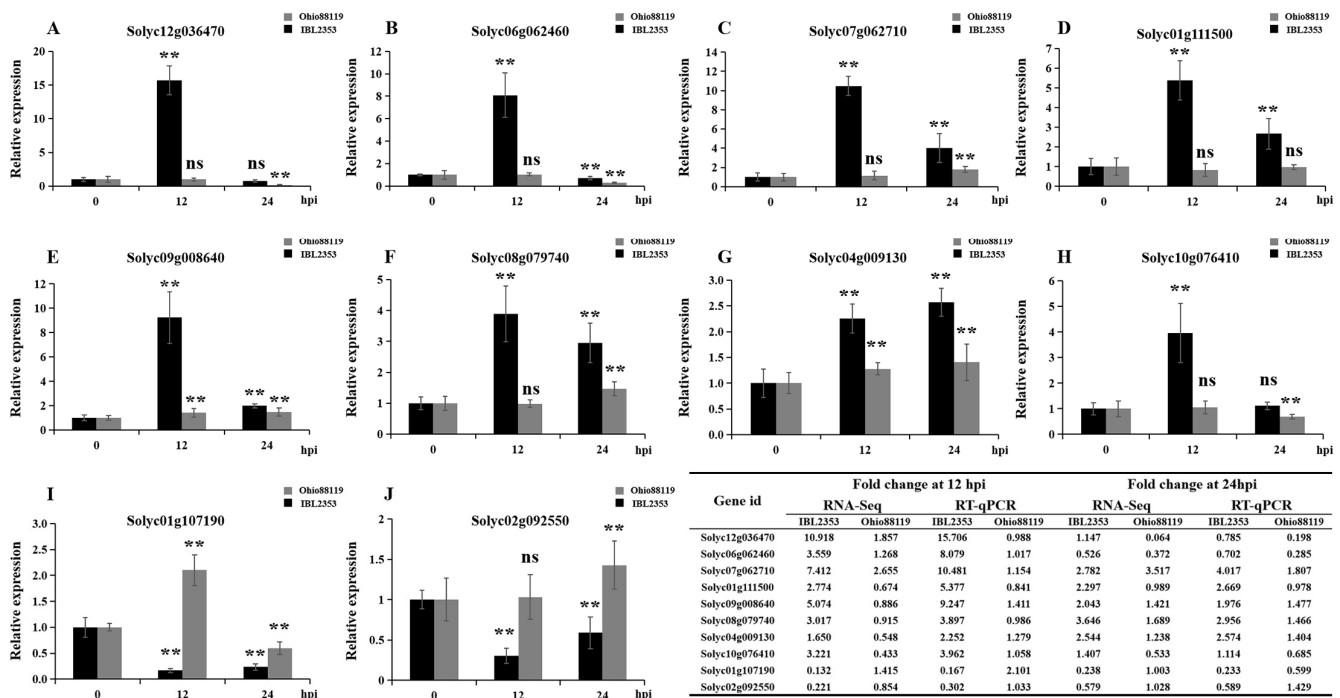

**Figure 7.** RT-qPCR analysis of 10 selected DEGs. (**A–J**) display the relative expression values of 10 selected genes in *Cm*-susceptible and *Cm*-resistant tomato lines at two time-points, respectively. The transcript expression level of each gene at 0 hpi was set as 1.0. ** denotes very significant differences compared to 0 hpi based on independent-samples *t*-test ($p < 0.01$), ns denotes no significant differences compared to 0 hpi ($p > 0.05$). The table displays the fold change comparison between the RNA-Seq analysis and RT-qPCR validation.

### 2.7. Silencing of WAKL20 Enhanced Susceptibility to Cm

Of the 25 genes associated with the defense response identified above (Figure 4), *WAKL20* (Solyc09g008640) was selected for analysis because it belongs to the WAKS sub-family that plays a critical role in innate resistance to multiple pathogens in different

crops [21–23]. RNA sequencing and RT-qPCR data show this gene is significantly up-regulated in IBL2353, but hardly changed in the susceptible line (Figures 4 and 7). *WAKL20* was the only member of the WAKs subfamily that was significantly induced and the expression changed in IBL2353 (Figure S2). To further explore whether *WAKL20* plays a positive role in resistance to *Cm* infection, we performed virus-induced gene silencing (VIGS) experiments in IBL2353. The leaves of positive control *CaPDS*-silenced plants bleached rapidly whereas the empty vector plants stayed green (Figure 8A,B). In addition, the *WAKL20*-silenced plants withered upon further incubation (Figure 8C–E). As shown in Figure 8F, the bands of pTRV1 and pTRV2 can be detected in the control plant and the three silenced plants. The expression level of *WAKL20* in the control plants (pTRV2) and silenced plants (pTRV1 + pTRV2-*WAKL20*) was identified using RT-qPCR, which showed the silence efficiency of three representative plants was 67.2%, 77.5%, and 82.1%, respectively (Figure 8G). The enhanced susceptibility to *Cm* of the *WAKL20*-silenced plants was also revealed by counting interior bacterial population in the stem, as shown by the significant increase in the *Cm* bacteria number compared to the control plants (Figure 8H). These results showed the silencing of *WAKL20* in IBL2353 resulted in the disease susceptibility to the *Cm* infection.

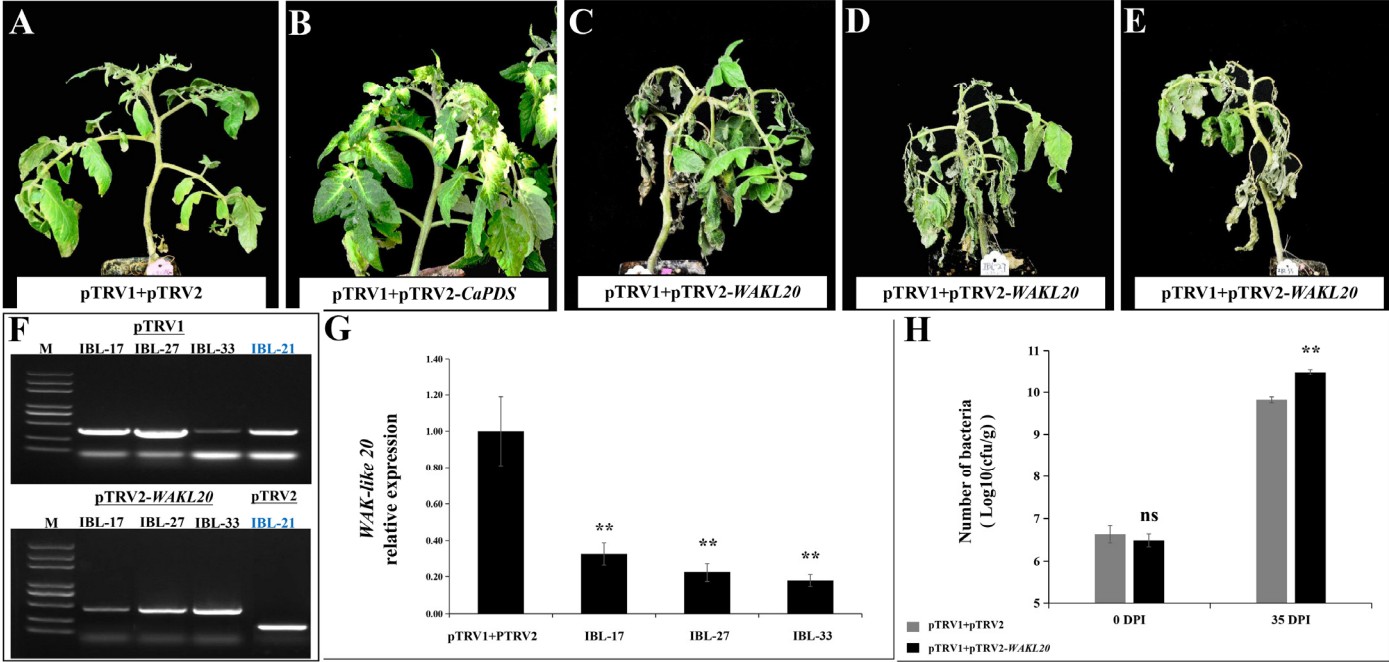

**Figure 8.** Silencing of *WAKL20* in tomato plants led to disease susceptibility to the *Cm* infection. (**A**) Phenotype of the empty vector plant (IBL-21) at 35 dpi. (**B**) The positive control *CaPDS*-silenced plants were light-bleached in leaves after 30 days. (**C–E**) The disease symptoms of three *WAKL20*-silenced plants at 35 dpi (IBL-17, IBL-27, and IBL-33). (**F**) PCR-based detection of pTRV1 and pTRV2-*WAKL20* plasmids in three silenced plants and negative control pTRV2 plant. (**G**) Relative expression of *WAKL20* in leaves collected from injected plants with pTRV1 + pTRV2 and pTRV1 + pTRV2-*WAKL20* (IBL-17, IBL-27, and IBL-33), respectively. Significant differences were determined using independent-samples *t*-test ($p < 0.01$). ** denotes very significant differences compared to the control plants based on independent-samples *t*-test ($p < 0.01$). (**H**) Bacterial growth in *WAKL20*-silenced plants at 35 dpi. ** denotes very significant differences between the control and *WAKL20*-silenced plants using independent-samples *t*-test ($p < 0.01$), ns denotes no significant differences between the control and *WAKL20*-silenced plants ($p > 0.05$).

## 3. Discussion

It has been more than 110 years since bacterial canker in tomato was first reported [4]. There is, as yet, no commercial variety with considerable resistance against the pathogen [3,8,12].

Up to now, *S. habrochaites* LA407 and *S. arcanum* LA2157 have been two of the most studied *Cm*-resistant wild accessions; the resistance-related characteristics of them have been reported [6,20,24–26]. The interaction of the *Cm* pathogen with tomato hosts was carried out with systemic infection through the vascular system, and the infection characteristics were the necrosis and cankers on the stems, unilateral or whole-plant wilting, and the petioles of the susceptible plants (Figure 1). These results suggest that there is an effective and complex immune response after *Cm* pathogen invasion in the resistant tomato. Thus, RNA-Seq transcriptome profiling was performed to unravel the molecular elements of this interaction. In tomato, three RNA-seq analyses detected the DEGs after *Cm* infection [13–15]. One study only used the susceptible cultivar as the sequencing object [13], and another two both used a comparative analysis between the cultivar and the wild LA2157 [14,15]. Because of a lack of a good reference genome sequence for wild LA2157, the average mapping percentages of LA2157 libraries were relatively poor: 85–90% and 77.3%, respectively [14,15]. In addition, the greater genetic background difference between the wild species and cultivars induced genetic background noise. In this study, we used one inbred backcross line, IBL2353, as the target of comparative analysis, which improved the mapping percentages (high to 95.0%) and large genetic variation with cultivars based on the high DEGs shared ratio (Figure 2). The same DEGs in the two present materials accounts for a large proportion of the total DEGs, achieving 50% and 32% at 12 hpi and 24 hpi, respectively (Figure 2C,D); therefore, the analysis and screen range of candidate genes were narrowed to a smaller gene amount.

This paper focuses on those unshared DEGs between the resistant and susceptible line. In this study, three E3 Ubiquitin-protein ligase genes (Figure 4) and two ubiquitin-protein transferase activator genes (Solyc06g072830 and Solyc08g005420) were only up-regulated in resistant IBL2353 after *Cm* infection (Tables S3 and S4). It is known that the Ubiquitin-26S proteasome system (UPS) is one of the critical pathways in plants controlling plant immunity [27] and regulating the accumulation of intracellular nucleotide-binding leucine-rich repeat (NB-LRR) immune receptors [28,29]. Many studies have indicated that E3 Ubiquitin Ligases play crucial roles during plant–pathogen interactions, such as thickening the cell wall, promoting the accumulation of $H_2O_2$ at the infected site [30], degrading the virulence factors of pathogens to protect host plants [31], and participating in pathogen-associated molecular PTI [32]. Additionally, in tomato, a U-box E3 ligase protein 24 (*SlPUB24*) has been reported as a positive regulator of bacterial spot resistance by influencing SA content, PR1 and NPR1 expression, and cell wall reinforcement to prevent bacterial migration [33]. Future research should focus on functional validation experiments to verify whether the ubiquitylation was also involved the resistant response to *Cm* infection in tomato.

Plants have another natural immunity system which can be termed as effector-triggered immunity (ETI). In the ETI system, the defense response is triggered via the recognition of pathogen-derived effectors by the resistant *R* gene products [34]. Several reported *R* genes or resistant-related genes were also up-regulated significantly in the resistant line, such as genes coding a probable CAF1 homolog 11 and NHL protein 6, two putative disease resistance RPP13-like protein 3, and an abscisic acid receptor PYL4-like (Figure 4). CAF1 in *Solanaceae* plants was positively related to pathogen resistance; for example, the overexpression of the CAF1 protein in tomato plants contributed to enhanced resistance to the pathogen *Phytophthora infestans*, and VIGS of *CaCAF1* in pepper plants led to an enhanced susceptibility to the pepper bacterial spot pathogen *Xanthomonas axonopodis* pv. *vesicatoria* [35]. CsCAF1 deadenylase activity was also reported to contribute to citrus canker resistance, possibly by regulating the transcription or stability of susceptibility genes *CsLOB1* [36]. *StPOTHR1*, a member of the NHL gene family, was specifically up-regulated in inoculation sites and promoted resistance against *Phytophthora infestans* in potato by inhibiting the rapid proliferation of pathogens [37]. RPP13 is a known resistance (*R*) gene, conferring resistance to wheat powdery mildew and Arabidopsis downy mildew [38,39]. The *PYL4* gene is an essential upstream regulator in the ABA signaling pathway and the silencing of *VvPYL4* reduced the expression of ABA and JA signaling pathway related genes

and grapevine resistance to downy mildew [40]. These results provide indications that the above *R* genes probably play roles in the defense response to *Cm* infection in tomato.

TFs that directly regulate the defense-associated gene expression play an important part in plant immunity, including in tomato [6,13,41]. Here, 57 TFs specially involved in the resistance response to *Cm* infection and up-regulated TFs mainly belong to three categories (Figure 5A), including the R2R3-MYB, bHLH, and bZIPs. The top up-regulated type was R2R3-MYB TFs. They control a wide variety of processes, including phenyl-propanoid metabolism and secondary cell wall formation [42]. MYB44 was reported to promote resistance to eggplant bacterial wilt via facilitating the expression of spermidine synthase [43], and, herein, the MYB44 (Solyc02g092930) expression increased in IBL2353 after *Cm* infection. Some bHLH proteins have been shown to confer resistance against various pathogens in various crops, such as improving resistance against *Phytophthora sojae* in Glycine max [44], improving resistance against *Xanthomonas oryzae* pv. *oryzae* in rice [45], and improving powdery mildew resistance in tobacco [46]. In citrus canker, CsbZIP40 plays a positive role in pathogen response and tolerance along with the SA signal pathway [47]. The overexpression of *CabZIP1* and *CabZIP2*, respectively, in the transgenic Arabidopsis plants confer enhanced resistance to *Pseudomonas syringae* pv. *tomato* DC3000, and *CabZIP2*-silenced pepper plants are susceptible to infection by the virulent strain of *X. campestris* pv. *Vesicatoria* [48,49]. The tomato orthologues of the above-mentioned three TF families were induced by *Cm* infection, suggesting that the functions of these TFs in disease response are highly conserved in a wide range of plants.

TFs may also act as transcriptional repressors. WRKY and AP2/ERF-ERF were top two types of the significantly down-regulated TFs (Figure 5B). WRKYs are one of the largest families of transcriptional regulators in plants, and most WRKYs are negative regulators of plant immunity [50–52]. In chili pepper leaves, VIGS of *CaWRKY1* resulted in the reduced growth of *Xanthomonas axonopodis* pv. *vesicatoria* race 1, whereas overexpressing transgenic plants displayed accelerated hypersensitive cell death responding to *Pseudomonas syringe* pv. *Tabaci* and tobacco mosaic virus [53]. CaWRKY40b in pepper plays a negative role in response to *Ralstonia solanacearum* by directly regulating defense genes [54]. Another of the top down-regulated types, AP2/ERF-ERF, is also one major TF category involved in defense pathways with critical roles in immune responses in plants [55]. In tomato, a pathogen inoculation assay revealed that *SlERF84* negatively regulates the plant defense response to *Pseudomonas syringae* pv. *tomato* DC3000 [56]. Earlier studies showed that ethylene-insensitive *Nr* plants and ethylene synthesis mutants inoculated with *Cm* pathogen showed a delayed onset of disease symptoms (by several days) and less serious wilting, compared with wild-type plants in tomato [57]. These results suggested that ethylene is a major hormone signal in the response to *Cm* infection and regulates disease progression in the tomato hosts.

The studies on susceptibility genes were gradually increased because the inactivation of *S* genes could lead to obtaining durable and broad-spectrum resistance compared with the *R* genes in crops, and *S* genes are usually conserved among plant species [18,19]. In this analysis, we screened eight *S* gene homologs from 28 reported *S* genes specially regulated in the susceptible line (Figure 6, Table S7). BIK1 and LOB genes accounts for half of them. BIK1 was proposed to cause rice leaf blight disease susceptibility by interacting with the effector XopR of the *Xanthomonas oryzae* pv. *oryzae* [58]. LOB has been also verified to function in *Arabidopsis Fusarium* wilt susceptibility [59] and citrus bacterial canker disease susceptibility [60,61] with the involvement of JA signaling. We have been performing transient silencing experiments of LOB in the susceptible line, and found that the Ohio88119-silenced plants show less susceptibility symptoms and bacterial population in stems compared to the control plants (Y.W. unpublished data). The function of these eight *S* gene homologs in response to *Cm* infection needs to be verified by further silencing or knockout experiments in susceptible cultivars in the future.

RLKs play a critical part in PTI against diverse pathogens in plants [6,9]. Our data showed more RLKs in the resistant line induced by *Cm* infection than in the susceptible line,

and among them, the WAK(L)s are a subfamily with a cell-wall-associated galacturonan-binding domain, which are the only known proteins to act as a direct link between the plasma membrane and the cell wall [62]. The WAK(L)s play a critical part in innate resistance to multiple pathogens through increasing cellulose and phytoalexin synthesis to bolster cell wall integrity and up-regulating specific pathogen defense genes in different crops [22–24]. In tomato, *SlWak1* has been discovered to be essential for apoplastic immune responses to *Pseudomonas syringae* pv. *tomato* through the cooperation with Fls2/Fls3, accompanied by the regulation of callose deposition [22]. Moreover, *CsWAKL08* has also been characterized to make a contribution to the resistance against bacterial canker by mediating ROS homeostasis and activating JA signaling in citrus [23]. The overexpression of the *OsWAK25* gene in rice promoted resistance to the hemi-biotrophic pathogens *Xanthomonas oryzae pv. oryzae* (*Xoo*) and *Magnaporthe oryzae* [24]. The expression level of *WAKL20* was significantly increased at 12 hpi in IBL2353 (Figures 4 and 7) when combining the data of RNA-seq and RT-qPCR, and this result was consistent with the up-regulated expression of *WAKL20* in the resistant line LA2157 at 8 hpi after *Cm* infection in a previous study [15]. In contrast, the weaker induction of *WAKL20* in the susceptible line led to the compromised manifestation of the sensitive reaction. Moreover, the transient silencing of the *WAKL20* gene in IBL2353 plants exhibited increased susceptibility to *Cm* infection (Figure 8). These results indicated that *WAKL20* plays an important role against the attack of *Cm* in tomato and may participate in the PTI response. In future, stable genetic transformation via the knocking out or a complementary assay of *WAKL20* is needed to verify further the resistant function and illuminate its resistance mechanism related to cell wall signal response.

## 4. Materials and Methods

### 4.1. Plant Materials and Pathogenic Cm Strain

The resistant line IBL2353 and the susceptible processing tomato Ohio88119 were selected for the present study. IBL2353 is a tomato line bearing the *Rcm2.0* locus derived from an inbred backcross breeding program to introgress the partial resistance from LA407. The seeds were kindly provided by Dr. David M. Francis in the Department of Horticulture and Crop Science at Ohio State University. *Clavibacter michiganesis* pathogenic strain GS12102, an isolate collected from the Gansu Province of China, was kindly provided by Prof. Laixin Luo in the Department of Plant Pathology at China Agricultural University. It was identified as a highly pathogenic and toxic strain with toxicity testing.

### 4.2. Growth Conditions and Inoculation Treatments

All seeds were surface-sterilized with 75% alcohol for 1 min and 4% NaClO for 8 min, and then were sown in pots and kept under a 16 h/8 h photoperiod and $25 \pm 1\ °C$ in a growth chamber in the College of Horticulture, China Agricultural University. Bacteria of GS12102 strain stored at $-80\ °C$ were prepared in an LB agar plate to expand propagation for 48–72 h at 28 °C. Bacterial cells were gathered and the inoculum was prepared with sterile water containing 10 mM $MgSO_4 \cdot 7H_2O$. The tomato seedlings grew to 5–6 true leaf stages, and were inoculated with the bacterial suspensions ($3 \times 10^8$ CFU/mL, OD600 = 0.5) at the stem base with a cotyledonary node, with 10 mM $MgSO_4 \cdot 7H_2O$ as the mock inoculation. According to our pre-experiment data and previous research results [15,63], we chose 12 h and 24 h as the time-points. The plants were sampled (true leaves adjoining the inoculation site) at 0, 12, and 24 hpi, immediately immersed in liquid nitrogen, and stored at $-80\ °C$ until RNA isolation. The plants were maintained until finishing the investigation of disease symptoms.

### 4.3. RNA Extraction and Library Preparation for Illumina Sequencing

Total RNA was extracted from the leaf samples in the inoculated plants with a DP432-RNAprep pure plant kit (TIANGEN, Beijing, China) in accordance with the manufacturer's direction. The concentration and purity of RNA for sequencing were measured with NanoDrop 2000 (Thermo Fisher Scientific, Wilmington, DE, USA) and the integrity was

assessed with the RNA Nano 6000 Assay Kit of the Agilent Bioanalyzer 2100 system (Agilent Technologies, Santa Clara, CA, USA). The 18 samples (9 samples of each tomato line) were sent to Biomarker Technologies Co., Ltd. in Beijing for sequencing library construction. The libraries were generated with NEBNext UltraTM RNA Library Prep Kit for Illumina (NEB, Ipswich, MA, USA) according to the manufacturer's direction; meanwhile, index codes were augmented to attribute sequences to each sample. Different libraries were pooled according to the target amount of offline data. High-throughput sequencing was carried out using the Illumina NovaSeq 6000 platform, with 150 bp paired-end reads.

### 4.4. Transcriptome Data Processing

The raw reads were processed by using a bioinformatic pipeline tool, BMK Cloud (www.biocloud.net, accessed on 6 December 2022) online platform. First of all, raw reads in the fastq format were filtered to attain clean reads. Then, clean reads were mapped to the *S. lycopersicum reference genome ITAG4.0 with the HISAT2 soft version 2.0.4* (http://ccb.jhu.edu/software/hisat2/index.shtml, accessed on 8 March 2022) [64], and the assembly of the transcriptome was performed with StringTie version 2.2.1 (https://ccb.jhu.edu/software/stringtie/index.shtml, accessed on 10 March 2022) [65]. Gene function annotation was on the basis of KO (KEGG Ortholog database), Nr (NCBI non-redundant protein sequences), and GO (Gene Ontology). DEGs analysis was processed through the DESeq2 program package version 1.30.1 (http://www.bioconductor.org/packages/release/bioc/html/DESeq.html, accessed on 6 December 2022). Fragments per kilobase per million (FPKM) was applied to represent the normalized expression value and is displayed in Table S8. The *p* values were adjusted with the Benjamini and Hochberg approach for controlling the false discovery rate. The gene expression profiles at 0 hpi were used as the baselines; genes with an FDR < 0.01 and a fold change ≥2 were considered as differentially expressed. GO enrichment analysis of the DEGs was performed using the topGO version 2.48.0 based on Wallenius noncentral hypergeometric distribution [66]. Additionally, the KOBAS software (version 2.0) was used for DEGs enrichment in the KEGG pathways [67,68]. TF prediction was performed using the software iTAK version 1.0 [69]. The heat maps were obtained with the use of the pheatmap version 1.0.12, and the raw expression data were normalized separately based on the Z-score formula: (x-mean(x))/sd(x) in the heat maps.

### 4.5. Gene Expression Validation with RT-qPCR Analysis

The preparation of total RNA was same as the procedure in 4.3. First-strand cDNA was synthesized with 1 µg of total RNA using HiScript® II Q RT SuperMix for qPCR kit (Vazyme Biotech Co., Ltd., Nanjing, China) in accordance with the manufacturer's directions. Quantitative RT-PCR was performed in a 15 µL reaction with ChamQ SYBR qPCR Master Mix (Vazyme Biotech Co., Ltd.) according to the instructions. The primers are listed in Table S9. RT-qPCR was performed in 384-well plates on the 7500 real-time PCR system (Applied Biosystems, Foster City, CA, USA), with an initial denaturation step set at 95 °C for 30 s, followed by 40 cycles of denaturation and annealing at 95 °C for 10 s and 60 °C for 30 s, respectively. Relative expression values were calculated using the comparative *Ct* method ($2^{-\Delta\Delta Ct}$) with *EF-1α* (Solyc06g0050600) as the reference [70].

### 4.6. Virus-Induced Gene Silencing (VIGS)

A VIGS system based on the tobacco rattle virus (TRV) was applied to identify the gene function [71]. The CDS fragments of *WAKL20* (300 bp) and *CaPDS* (452 bp) were PCR-amplified with tomato cDNA and cloned into the pTRV2 vector to construct the plasmids pTRV2-*WAKL20* and pTRV2-*CaPDS*, respectively. pTRV1, pTRV2-*WAKL20*, pTRV2, and pTRV2-*CaPDS* were then mobilized into the *Agrobacterium tumefaciens* strain GV3101 via electroporation. *Agrobacterium tumefaciens* strains containing pTRV1 were mixed with that containing pTRV2-*WAKL20*, pTRV,2 and pTRV2-*CaPDS* in equal amounts, respectively. The resuspension of mixed bacterial strains was infiltrated into the cotyledons and one true leaf in two-leaf stage seedlings via syringe infiltration. Plants infiltrated with

pTRV1 + pTRV2-*CaPDS* and pTRV1 + pTRV2 were utilized as the positive and negative control, respectively [72]. The infiltrated plants were grown at 22 °C in 16/8 h light/dark in a culture room. When light bleaching occurred in the *CaPDS*-silenced plants (usually 2 weeks after inoculation), RT-PCR was performed to detect pTRV1 and pTRV2. Seedlings at the 5–6 true leaf stages were inoculated with the *Cm* (OD600 = 0.5), and then were maintained until 6 weeks to investigate the disease symptoms. The bacterial populations in the inoculated plants were determined using the dilution plate method [73]. The stem transection of *WAKL20*-silenced plants and control plants was used for testing the bacterial population at 35 dpi. The data were analyzed using the SPSS 23 software. At least 100 plants were infiltrated with pTRV1 + pTRV2-*WAKL20* and at least 30 plants were injected with plasmid mixtures of each control, respectively. The whole experiment was repeated twice. The primers used here are displayed in Supplementary Table S10.

## 5. Conclusions

In this study, high-throughput RNA sequencing was conducted to identify *C. michiganensis*-responsive genes in *Cm*-resistant and *Cm*-susceptible tomato lines. Upon *Cm* inoculation at 12 hpi and 24 hpi, special genes with different expression patterns between the resistant and susceptible lines were identified and annotated through GO terms and KEGG pathway analysis. From these DEGs, ten defense-responsive genes were selected and further validated for their different expression changes via RT-qPCR. Meanwhile, we also screened out eight S gene homologs specially induced in the susceptible line. In the end, *WAKL20* was identified to be specifically up-regulated in IBL2353 and validated as essential for the resistance response to *Cm* infection. Further research on other candidate genes is envisioned to expand the arsenal of candidate resistance and susceptibility genes in response to bacterial canker of tomato.

**Supplementary Materials:** The following supporting information can be downloaded at: https://www.mdpi.com/article/10.3390/horticulturae9020242/s1, Figure S1: Identification of DEGs in breeding line IBL2353 and *S. lycopersicum* cv. Ohio88119 following infection with *Cm*; Figure S2: Expression profiles of WAKs subfamily genes in two lines after *Cm* infection; Table S1: Statistics data from Illumina sequencing on two lines; Table S2: Total reads mapped to the *S. lycopersicum* reference genome ITAG4.0; Table S3: Enriched functional groups in biological process ontology (BP) in genes with different expression patterns between resistant and susceptible lines; Table S4: Enriched functional groups in molecular function ontology (MF) in genes with different expression patterns between resistant and susceptible lines; Table S5: Enriched functional groups in cellular component ontology (CC) in genes with different expression patterns between resistant and susceptible lines; Table S6: Enriched KEGG pathways in genes with different expression patterns between resistant and susceptible lines; Table S7: Reported susceptibility genes in different plant species; Table S8: The fragments per kilobase per million (FPKM) values for each unigene; Table S9: Primers for RT-qPCR; Table S10: Primers for VIGS vector construction and detection. References [74–97] are cited in the supplementary materials.

**Author Contributions:** Conceptualization, Y.W. and W.Y.; validation, S.D. and Z.L.; formal analysis, S.D.; investigation, S.D., Z.L. and X.L.; resources, Y.W.; writing—original draft preparation, S.D.; writing—review and editing, S.D. and Y.W.; supervision, Y.W.; funding acquisition, Y.W. All authors have read and agreed to the published version of the manuscript.

**Funding:** This research was funded by the National Natural Science Foundation of China, grant number 31501753.

**Data Availability Statement:** Raw and processed data are available from National Center for Biotechnology Information (NCBI) under the accession PRJNA 929084.

**Acknowledgments:** The authors are truly grateful to David M. Francis in the Department of Horticulture and Crop Science at Ohio State University for providing the seeds, and appreciate Laixin Luo in the Department of Plant Pathology at China Agricultural University for providing the bacterial pathogen strain GS12102.

**Conflicts of Interest:** The authors declare no conflict of interest.

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
