# Peer review of "Comparative Transcriptome Analysis Reveals Potential Genes Conferring Resistance or Susceptibility to Bacterial Canker in Tomato"

_horticulturae, doi:10.3390/horticulturae9020242_

Round 1

Reviewer 1 Report

The authors present a good manuscript focusing on the infection of a pathogenic strain in susceptible and resistant solanum lines. The working hypothesis must be clearly specified in the introduction, and the justification of why they worked on 12 and 24 hpi (hour post inoculation?) needs to be added to the material and methods

Line 17: on these Go terms, which were associated with susceptible or resistant lines? please specify'

Line 19: How were these genes screened out?

Line 41: what does the word accession refer to?

Line 447: use subscripts wherever necessary.

Line 451: Define HPI

Line 465: Libraries must be submitted to a public database

Section 4.4 add the version of each software used

Line 501: Specify why these genes were selected

Section 2.3: add quantitative data to specify such higher enrichments and/or genes

In figure 4, Why is there a red annotation?

In figure 7, add letters to subfigures.

Author Response

Reply to the Reviewer 1 

The authors present a good manuscript focusing on the infection of a pathogenic strain in susceptible and resistant solanum lines. The working hypothesis must be clearly specified in the introduction, and the justification of why they worked on 12 and 24 hpi (hour post inoculation?) needs to be added to the material and methods

Reply: Thank you for pointing out this issue. About working hypothesis, in Line 49-53 and Line 74-78 of the introduction, we specified it in the manuscript revised version.

About justification of why we worked on 12 and 24 hpi, we have added the base of time-points choice into “material and methods” (Line 456). The choice was based on research foundations of two aspect. One is the data of our pre-experiment on PR-1 gene expression in our line (Figure R1). It’s well known that plant pathogenesis-related protein 1 (PR-1) are among the most abundantly produced proteins in plants on pathogen attack, and PR-1 gene expression has long been used as a marker for salicylic acid-mediated disease resistance. The study also showed the resistance to bacterial canker was salicylic acid-mediated disease resistance (Reference 13). Our result showed the PR1 gene expression was induced to the highest level at 12 dpi after Cm infection in Ohio88119 (Reply Figure1). Meanwhile, the similar study in 2021 were also chose the 8 - 24 h after inoculation (Reference 15). Another research foundation was the proteomic analysis data, and the study found the protein regulation of defense-related genes were detected early at 72 h after infection in the line IBL2353 containing Rcm 2.0 (new Reference 63). This result suggested us gene expression time be earlier than 72 h of proteins translation based on the central dogma in molecular biology. Based on above justifications, we chose the 12 and 24 h as the work time points in this study.

  1. Coaker, G.L.; Willard, B.; Kinter, M.; Stockinger, E.J.; Francis, D.M. Proteomic analysis of resistance mediated by Rcm 2.0 and Rcm 5.1, two Loci controlling resistance to bacterial canker of tomato.Mol. Plant Microbe Interact. 2004, 17, 1019-1028.

Figure R1. Expression profiles of PR1 gene after inoculation with Cm in Ohio88119.

Line 17: on these Go terms, which were associated with susceptible or resistant lines? please specify'

Reply: Sorry it made you wondering. Due to the word limit of Abstract, in this line 74 we didn’t specify and detail them. About reply to this question, we described them in Line 142-143 (“Results” Section 2.3). We only chose those DEGs with different expression patterns between two lines (means at same time-point, genes up-regulated in resistant line but non-changed or down-regulated in susceptible line and genes down-regulated in resistant line but non-changed or up-regulated in susceptible line) were used for GO terms analysis. So, there was no GO terms only associated with one susceptible or one resistant line.

Line 19: How were these genes screened out?

Reply: Sorry it seems unclear. But in “Results” Section 2.4, the first sentence (Line 176-177) described as following “Combining the DEGs values (LogFC ≥ 1) with the GO term and KEGG pathway enrichment analysis and the gene functional annotation associated with resistance”. That means that those genes significantly induced in resistant line during 24 hours, combined with previous reports or functional annotation of NCBI database on genes associated with resistance, at the same time belong to the top terms or pathway in GO term and KEGG pathway enrichment analysis in present study just were screened out. The gene number satisfy above three conditions was 25, and were showed in Figure 4.

Line 41: what does the word accession refer to?

Reply: The word “accession” is a common term in “Genetics and Breeding”, includes wild species and cultivars and breeding lines.

Line 447: use subscripts wherever necessary.

Reply: Thank you for pointing out this carelessness. We have changed it into MgSO4·7H2O includes other two places in manuscript.

Line 451: Define HPI

Reply: Thank you. We have added its definition as “hours post inoculation” in the first appearance Line 121.

Line 465: Libraries must be submitted to a public database

Reply: We agreed with you. The manuscript wrote the database name “Solanaceae Genomics Network” in last part (“Data Availability Statement”), and it’s a common Solanaceae public database. Once the manuscript is accepted, we’ll upload the libraries to database.

Section 4.4 add the version of each software used

Reply: Thank you for your suggestion. We have added all software versions in Section 4.4.

Line 501: Specify why these genes were selected

Reply: The first gene “pTRV2-WAKL20” is our target gene (same with the title of Section 2.7), the second gene “pTRV2-CaPDS” is the positive control gene (means bleached CaPDS-silenced plant in leaves can prove our silence experiment system work well). “TRV1 and TRV2” is the two constitutive genes on two vectors used for virus- induced gene silencing experiments.

Section 2.3: add quantitative data to specify such higher enrichments and/or genes

Reply: Thank you for pointing out this issue. In this section, we chose the most enrichment intensity 20 terms rather than the top quantity 20 terms based on the raw data. Because the analysis we made here was “GO Term Enrichment analysis” other than “GO Term analysis”. That means the higher quantitative data of term genes did not account for higher enrichment intensity. A lower p value indicates greater enrichment intensity (Figure 3 legend). Therefore, about the raw quantitative data of the highest 20 terms in Figure 3, we showed them only in the Supplement Table S3-S6. Now, the top 20 significant enriched terms have been displayed with bold font in Table S3-S6, and the raw quantitative data of genes in each GO term were shown in column C of Table S3-S6.

In figure 4, Why is there a red annotation?

Reply: Thank you for pointing out this carelessness. In the compressed files of Figures uploaded separately, we had removed the red color on last gene names, but we forgot amended it in original manuscript. Now, we have removed the red annotation in Figure 4.

In figure 7, add letters to subfigures.

Reply: Thank you for pointing out this issue. We have added letters “A-J” to subfigures in Figure 7.

Reviewer 2 Report

Dear edithor

I am writing about the manuscript “Comparative Transcriptome Analysis Reveals Potential Genes Conferring Resistance or Susceptibility to Bacterial Canker in Tomato” by Deng et al.  As it is referred to in the title, the manuscript analyses the transcriptome of two lines of Solanum lycopersicum challenge with the pathogenic bacteria Clavibacter michiganensis, which one of the lines is susceptible, and the other is resistant. The authors refer that both lines are close genomically and allow observing differences in the genomic expression avoiding the noise caused by more extensive differences. . The works report two sets of genes differentially expressed, one in the resistant line and the other in the susceptible line, pointing out interesting genes involved in natural resistance susceptible to be investigated.

In my opinion, this is an interesting, well-written, and presented work that could be published after addressing some minor problems.

My principal criticism is the lack of information about the origin of the S. lycopersicum lines. This information (references included) is missing in the introduction, and also in M&M section. While Abstract provided some comments about the line's origin, it would be important to provide this information in Induduction and M&M.

Particular comments:

Line 17. Please define GO.

Line 74. Please provide the context of the line IBL2353, its characteristics, and references where it is described to understand what is the IBL2353.

Line 101. Please indicate in each case (panel), which of the plants were inoculated with mock and which were inoculated with Cm.

Line 115. Please clarify. Does it mean differential expressed genes of one line compared with the other? or Does it means DEGs of one time compared with initial time?

Line 173. Please define FC.

Line 220. Please provide table S7, it is missing in the supplementary material.

Figure 7. Please review the scales of the Y axes of the Solyc04g009130 and Solyc01g107190 graphs.

Line 265. Please add "Solyc09g008640" after “WAKL20” to do easier the relation with Figure 7.

Line 432. "4.1. Plant Materials and pathogenic Cm Strain". Due to the work approach, it is very important to have more information about the IBL23 and Ohio88119  Lines. Please provide the references where both lines are described.  Particularly it is interesting to know which is the relation between both lines. Is Ohio88119 the parental of  IBL23? Are they not related? Why are they especially close? How was the inbred backcross breeding performed? How was the Rcm2.0 locus mobilized to  IBL23? Is there more information about the Rcm2.0 locus? Please provide the information.

It is not so crucial, but also important, the characterization of Clavibacter michiganesis. Please provide the reference where it is described. How were the bacteria identified? Was its 16S sequence deposited? Please provide the information.

Author Response

Reply to the Reviewer 2 

My principal criticism is the lack of information about the origin of the S. lycopersicum lines. This information (references included) is missing in the introduction, and also in M&M section. While abstract provided some comments about the line's origin, it would be important to provide this information in Induction and M&M.

Reply: Thank you for this suggestion. We have added some information about the IBL 2353 line's origin in Induction (Line74-78), and not repeated in M&M. The origin of another line Ohio88119 was mentioned in original manuscript M&M (Line 437-441). Because it is only a common cultivar (S. lycopersicum) breeding by David M. Francis’s lab at the Ohio State University, and then the origin not be added again in Induction in view of conciseness.

Particular comments:

Line 17. Please define GO.

Reply: Thank you for pointing out this carelessness. We have changed it into “Gene Ontology” (Line 17).

Line 74. Please provide the context of the line IBL2353, its characteristics, and references where it is described to understand what is the IBL2353.

Reply: Thank you for pointing out this issue. Information about the line have been added (Line74-78). The original reference 25 described the characteristics of IBL2353, we have changed its order into Line 78 as new order 20.

Line 101. Please indicate in each case (panel), which of the plants were inoculated with mock and which were inoculated with Cm.

Reply: Thank you for pointing out this issue. We have added the inoculation or mock information in the legend of Figure 1 (Line 105-110).

Line 115. Please clarify. Does it mean differential expressed genes of one line compared with the other? or Does it means DEGs of one time compared with initial time?

Reply: Sorry, the DEGs in this line “number and overlap quantity of DEGs” means those DEGs of one time-point compared with initial time in same one line. We mentioned it in the legend of Supplementary Figure 1.

Line 173. Please define FC.

Reply: Thank you. We have changed it into “Fold change (FC)” (Line 176). 

Line 220. Please provide table S7, it is missing in the supplementary material.

Reply: Thank you for pointing out this carelessness. We have added it into supplementary material.

Figure 7. Please review the scales of the Y axes of the Solyc04g009130 and Solyc01g107190 graphs.

Reply: Thank you for pointing out these two mistakes. We have corrected the scales of the Y axes of the Solyc04g009130 and Solyc01g107190 graphs in Figure 7.

Line 265. Please add "Solyc09g008640" after “WAKL20” to do easier the relation with Figure 7.

Reply: Thank you for this suggestion. We have added "Solyc09g008640" after“WAKL20”in line 269, then readers can found this gene more easily in the Figure 7.

Line 432. "4.1. Plant Materials and pathogenic Cm Strain". Due to the work approach, it is very important to have more information about the IBL23 and Ohio88119 Lines. Please provide the references where both lines are described.  Particularly it is interesting to know which is the relation between both lines. Is Ohio88119 the parental of IBL23? Are they not related? Why are they especially close? How was the inbred backcross breeding performed? How was the Rcm2.0 locus mobilized to IBL23? Is there more information about the Rcm2.0 locus? Please provide the information.

It is not so crucial, but also important, the characterization of Clavibacter michiganesis. Please provide the reference where it is described. How were the bacteria identified? Was its 16S sequence deposited? Please provide the information.

Reply: Thank you for pointing out this issue. About the references where IBL2353 line were described have been replied in the question on Line 74 (in third reply). Ohio 88119 was not the parental of IBL2353. Both of them were breed by the same research group (David M. Francis’s lab) at the Ohio State University. Ohio 88119 is a processing tomato cultivar, and IBL2353 were developed through five generations backcrossing with S. habrochaites LA407 used a processing tomato cultivar breeding from their lab and then five generations selfing (current Reference 20). That group had made the genotypic analysis on IBL2353, and found a lower than expected proportion (4.2%) of the LA407 genome was recovered. How the Rcm2.0 locus mobilized to IBL2353 and more information about it have been detailed described in literature (Reference 12) and were cited in this study originally.

Another question on the characterization of Cm strain. This pathogen was provided by Prof. Laixin Luo in the Department of Plant Pathology at China Agricultural University (in Line 444 we added the Department name). And the original report about it was one Master Dissertation from them [R1], and the data about GS12102 was showed in Page 21. This bacterium was identified by amplifying the typical virulence genes celA and pat-1 on two plasmids pCM1 and pCM2 rather than characterizing the 16S sequence, and inoculating the tomato hosts for pathogenicity investigation. The amplification data was showed here (Figure R1), and the pathogenicity results were showed in manuscript Figure 1 and in Figure 1 and 2 (Reference 3) respectively (Horticulturae, Wang et al., 2022). Because the Cm strain GS12102 was secondly used in study and the experiment data of first utilization was already published in Horticulturae journal, so we only provide the reference in manuscript rather than more detailed information.

R1.  LV, Q. Detection of Clavibacter michiganensis subsp. michiganensis in tomato seed by using loop-mediated isothermal amplification. Master, China Agricultural University, Beijing, 2014.

A

B

Figure R1. The gel electrophoresis image from Cm strain GS12102. (A) The celA gene on pCM1 can be amplified in the strain GS12102. (B) The pat-1 gene on pCM2 can be amplified in the strain GS12102. Positive control was the Cm reference strain NCPPB382.

Round 2

Reviewer 1 Report

the authors improve the manuscript presentation. Acceptance is recommended after the authors submit the raw data to a public database 

Author Response

the authors improve the manuscript presentation.

Reply:Thank you for your suggestion. We asked one native English speaker to check  this manuscript, and revised his most revisions. And all these changes were tracked in this resubmitted version.